# SARS-CoV-2 m-RNA Vaccine Response in Immunocompromised Patients: A Monocentric Study Comparing Cancer, People Living with HIV, Hematopoietic Stem Cell Transplant Patients and Lung Transplant Recipients

**DOI:** 10.3390/vaccines11081284

**Published:** 2023-07-26

**Authors:** Natacha Bordry, Anne-Claire Mamez, Chiara Fedeli, Chloé Cantero, Cyril Jaksic, Pilar Ustero Alonso, Caroline Rayroux, Gregory Berra, Vera Portillo, Maeva Puntel, Sabine Yerly, Sébastien Bugeia, Garance Gutknecht, Mariagrazia Di Marco, Nicolas Mach, Paola Marina Soccal, Yves Chalandon, Alexandra Calmy, Alfredo Addeo

**Affiliations:** 1Department of Oncology, Geneva University Hospitals, University of Geneva and Swiss Cancer Center Leman, 1205 Geneva, Switzerland; natacha.bordry@hcuge.ch (N.B.);; 2Department of Haematology, Geneva University Hospitals and Faculty of Medicine University of Geneva, 1205 Geneva, Switzerland; anne-claire.mamez@hcuge.ch (A.-C.M.);; 3Department of Infectious Diseases, Geneva University Hospitals, University of Geneva, 1205 Geneva, Switzerlandalexandra.calmy@hcuge.ch (A.C.); 4Department of Pneumology, Geneva University Hospitals, University of Geneva, 1205 Geneva, Switzerland; 5CRC & Division of Clinical Epidemiology, Department of Health and Community Medicine, University of Geneva and Geneva University Hospital, 1205 Geneva, Switzerland; 6Laboratory of Virology, Division of Laboratory Medicine, Geneva University Hospitals & Faculty of Medicine, 1205 Geneva, Switzerland

**Keywords:** mRNA vaccines, SARS-CoV-2, immunodeficiency

## Abstract

Immunocompromised patients (ICPs) have a higher risk of developing severe forms of COVID-19 and experience a higher burden of complications and mortality than the general population. However, recent studies have suggested that the antibody response to SARS-CoV-2 mRNA vaccines could be highly variable among different ICPs. Using a collaborative, monocentric, prospective cohort study, we assessed anti-SARS-CoV-2 spike protein antibody titers following two and three doses of mRNA vaccines in four groups of ICPs (cancer [*n* = 232]: hematopoietic stem cell transplant [HSCT; *n* = 126] patients; people living with HIV [PLWH; *n* = 131]; and lung transplant [LT; *n* = 39] recipients) treated at Geneva University Hospitals; and healthy individuals (*n* = 49). After primo-vaccination, the highest anti-S antibody geometric mean titer (IU/mL) was observed in healthy individuals (2417 IU/mL [95% CI: 2327–2500]), the PLWH group (2024 IU/mL [95% CI:1854–2209]) and patients with cancer (840 IU/mL [95% CI: 625–1129]), whereas patients in the HSCT and LT groups had weaker antibody responses (198 IU/mL [95% CI: 108–361] and 7.3 IU/mL [95% CI: 2.5–22]). The booster dose conferred a high antibody response after 1 month in both PLWH (2500 IU/mL) and cancer patients (2386 IU/mL [95% CI: 2182–2500]), a moderate response in HSCT patients (521 IU/mL [95% CI: 306–885]) and a poor response in LT recipients (84 IU/mL [95% CI: 18–389]). Contemporary treatment with immunosuppressive drugs used in transplantation or chemotherapy was associated with a poor response to vaccination. Our findings confirmed the heterogeneity of the humoral response after mRNA vaccines among different ICPs and the need for personalized recommendations for each of these different groups.

## 1. Introduction

COVID-19 is a global public health issue with major societal trans-sectoral impacts. Since December 2019, COVID-19 has affected over 753 million people and is responsible for over 6.8 million deaths globally (as of 2 February 2023; https://COVID19.who.int/ [accessed on 20 July 2023]). Immunosuppression and immunodeficiency have been associated with an increased risk of severe COVID-19, requiring hospitalization, admission to the intensive care unit, and invasive ventilation and resulting in a higher risk of death [1].

Studies assessing the Pfizer-BioNTech BNT162b2 mRNA and Moderna mRNA-1273 vaccines showed that they have 95% efficacy in preventing severe COVID-19, but these trials systematically excluded immunocompromised patients (ICPs) [2,3].

Since then, several studies have been performed to investigate the immunogenicity of SARS-CoV-2 vaccines in immunocompromised populations. Compared to the general population, people living with HIV (PLWH) and patients with solid tumors showed similar seroconversion rates (90–95%) after two and three doses of mRNA vaccines [4,5,6,7]. However, patients receiving cytotoxic chemotherapy or monoclonal antibodies [4,5] had lower antibody titers when compared to patients not receiving these oncologic treatments [4,5]. By contrast, allogeneic hematopoietic stem cell (HSCT) patients and lung transplant (LT) recipients were less likely to develop a humoral immune response after mRNA vaccination [8,9,10]. A recent meta-analysis highlighted a poor seroconversion rate after COVID-19 vaccination in transplant patients, with only one-third achieving seroconversion after two doses [11].

These results suggest that the antibody response to SARS-CoV-2 mRNA vaccines was highly variable among different immunosuppressed patients. Less is known about the similarity and differences regarding the intensity and sustainability of the vaccine response and post-vaccine incidence and severity of COVID-19 manifestations after vaccination between these heterogeneous groups.

We report data from a collaborative monocentric, prospective cohort study assessing the short- and long-term humoral immune response (antibody titers), including COVID-19 incidence and severity, after two doses of mRNA vaccines (mRNA-1273 and BNT162b2) and a booster dose in cohorts of ICPs treated in a tertiary university hospital in Switzerland.

## 2. Results

### 2.1. Clinical Characteristics

The study cohort consisted of 577 patients, comprising four groups of ICP patients (*n* = 528) compared to healthy individuals (*n* = 49). The four cohorts include 232 cancer patients, 131 PLWH, 126 HSCT patients and 39 LT recipients (Table 1).

The median age of participants was 56 years (median age of healthy individuals, 30 years). Globally, 25% of the patient cohort was over 65 years old. Forty-three percent of patients had one or more comorbid conditions associated with a higher risk of developing severe COVID-19 (hypertension, diabetes, obesity, heart disease, pulmonary disease, tobacco use).

Patient characteristics are detailed in Appendix A. In summary, most cancer patients (Appendix A) had solid tumors (84%) and stage IV disease (60%). Approximately, one-third (29%) did not receive anti-cancer therapy within 6 months prior to SARS-CoV-2 vaccination. The most common anti-cancer therapy received was cytotoxic chemotherapy (35%) with alkylating agents (25%). Other treatments received 6 months prior to vaccination were surgery (14%), radiation (21%), hormonal therapy (13%), immunotherapy (13%) and target therapy (23%).

All PLWH (*n* = 131) were receiving antiretroviral therapy; 85% were virologically suppressed with undetectable HIV-RNA levels (<20 copies/mL). All had HIV-RNA levels below the threshold of 200 copies/mL and experienced immune restoration at the time of primo-vaccination (average CD4 T-cell count: 646.9 (±294.5) cells/mm^3^). The United States Centers for Disease Control and Prevention stage was A for 63.3% of participants, B for 19.5% and C for 17.2% (Appendix A).

In the HSCT group, the median time from transplant to first vaccine was 15 months (range: 6.7–27.3). Thirty-seven percent of patients had acute (2%) or chronic (28%) graft versus host disease (GVHD) at the time of vaccination. Thirty percent were under systemic immunosuppressive drugs at the time of the first vaccine dose (5% cyclosporine, 9% prednisone ≥ 10 mg per day, 11% tacrolimus, 2% mycophenolate mofetil [MMF] and 13% ruxolitinib). Six percent were receiving contemporary chemotherapy (either for active relapse or as maintenance) and five (4%) patients had received rituximab within 6 months. The median CD4 counts at vaccination were 226/mm^3^ (quartiles [Q1–Q3: 111–423) (Appendix A).

In the LT group, the etiology of the lung disease before transplant was chronic obstructive pulmonary disease (COPD) (46%), interstitial lung disease (23%), cystic fibrosis (18%), bronchiectasis (8%) and pulmonary hypertension (5%) (Appendix A). The medium time since the lung transplant was 85 months (quartiles [Q] 1–3: 28–133), with 13% having multiple organ transplants. Most patients were under triple immunosuppression (85%) and 28% had high doses of steroids during the 12 months prior to the first vaccine dose. The most common immunosuppressive drugs received were MMF and enteric-coated mycophenolate sodium (95%).

### 2.2. Short- and Long-Term Antibody Response after Two and Three SARS-CoV-2 mRNA Vaccine Doses

The antibody response measured based on anti-SARS-CoV-2 anti-S Ig titers and geometric mean titers (GMT) for the five groups after primo and booster vaccination is presented in Figure 1 and Table 2, respectively.

At baseline (before vaccination), 20.9% (*n* = 58) of cohort patients had positive anti-S Ig titers. One month (M1) after primo-vaccination (first 2 doses of vaccine]), the highest anti-S antibody GMT (IU/mL) was observed in healthy individuals (2417 IU/mL [95% CI: 2327–2500]), followed by the PLWH group (2024 IU/mL [95% CI: 1854–2209]) and cancer patients (840 IU/mL [95% CI: 625–1129], whereas the HSCT and LT groups had a weaker antibody response (198 IU/mL [95% CI: 108–361] and 7.3 IU/mL [95% CI: 2.5–22], respectively) (Figure 1 and Table 2).

From three to five months after primo-vaccination (M3-M5), the anti–S antibody titer decreased in healthy individuals (1812 IU/mL [95% CI: 1570–2092]) and the PLWH group (1177 IU/mL [95% CI: 1003 to 1380]), but remained stable in cancer patients (830 IU/mL [95% CI: 636–1082]), LT recipients (20 IU/mL [95% CI: 6.5–65]) and HSCT patients (388 IU/mL [95% CI: 111–1358]. Notably, in the HSCT group, only 18/126 patients had positive antibody titers 3–5 months after two vaccine doses. Indeed, most patients had already received a booster dose before reaching this time point.

The booster dose conferred a high antibody response after 1 month (booster_M1) in both the PLWH group (2500 IU/mL) and in cancer patients (2386 IU/mL [95% CI: 2182–2500]), a moderate response in HSCT patients (521 IU/mL [95% CI: 306–885]), and a poor response in LT recipients (84 IU/mL [95% CI: 18–389]). However, in the latter two groups, antibody titers showed an increase of 2 and 12 times, respectively, after the booster dose compared to the second dose. When available, long-term follow-up (at 6 months; booster_M6) in PLWH and HSCT patients after the booster showed stable antibody levels in both groups (2379 IU/mL [95% CI: 2240–2500] and 480 IU/mL [95% CI: 137 to 1676], respectively).

### 2.3. Proportion of Immunocompromised Patients with >300 IU/mL of SARS-CoV-2 Antibody after Two and Three Vaccine Doses

Based on the recommended guidelines (Swiss Federal Office of Public Health: https://www.infovac.ch/docs/public/coronavirus/bag/annexe1-COVID-19-recommandation-pour-vaccins-arnm-e--tat-210721.pdf [accessed 20 July 2023]) and published data [12], patients and healthy individuals were separated as “strong” or “poor” responders, based on Ig anti-S antibody titer, with a cut-off of 300 IU/mL (Figure 2). Between August and September 2021 in Switzerland, only poor responders were candidates for the booster dose. The proportion of poor responders was highly variable between the five groups. After primo-vaccination (M1), more than 84% and 38% of LT and HSCT patients, respectively, were poor responders compared to only 15% of cancer patients and <2% in the PLWH group. After the booster dose, we observed a higher proportion of patients reaching more than 300 IU/mL in the LT (48%) and HSCT (70%) groups.

### 2.4. Proportion of Immunocompromised Patients with SARS-CoV-2 Infection

The incidence of COVID-19 during follow-up is presented in Figure 3. Among all COVID-19 diagnoses, 62% were identified using a polymerase chain reaction (PCR) test and 38% were based on the new detection of anti-N antibody during follow-up, revealing an important number of asymptomatic or poorly symptomatic SARS-CoV-2 infections. SARS-CoV-2 variant determination was not performed. Most COVID-19 infections (69%) were diagnosed after January 2022 when Omicron variants became predominant in our setting.

After primo-vaccination, 31 (5%) patients contracted COVID-19 infection, mainly in the HSCT (*n* = 17) group and healthy individuals (*n* = 6). One month after the booster dose, 32 additional patients developed COVID-19, with a further 67 patients during the period from 1 to 6 months after the booster dose (data not available for LT recipients and cancer patients). No patient received prophylactic monoclonal SARS-CoV-2 antibody during the follow-up period. Only a few patients required hospitalization due to COVID-19 infection (HSCT, 14; LT, 3); none required intensive care unit admission. No death related to COVID-19 infection was reported.

### 2.5. Response Factors according to Immunosuppression Group

Finally, we tested the association between different predictors (e.g., immunosuppressive treatments, patients’ immune status, CDC tumor stage) and anti-SARS-CoV-2 anti-S Ig values > 300 IU/mL at any time point after vaccination (“strong responder”). In order to limit the scope of the analyses to immunosuppressed patients, healthy volunteers were excluded. Moreover, patients who already reached anti-SARS-CoV-2 anti-S Ig values > 300 IU/mL at baseline were also excluded. The odds ratios and their 95% CIs are reported in Table 3.

Immunosuppressive treatments during vaccination in the transplanted population (i.e., HSCT and LT patients), such as prednisone, tacrolimus, MMF and chemotherapy, were inversely associated with anti-SARS-CoV-2 anti-S Ig values > 300 IU/mL at any time point after vaccination, meaning that patients receiving these treatments were less likely to develop a “strong response”.

CD4 counts were measured in PLWH and HSCT patients at the time of the first vaccine dose. The average CD4 T-cell counts in the two groups were 646.9 (±294.5) cells/mm^3^ and 226 (111–423) cells/mm^3^, respectively. CD4 counts were positively associated with anti-SARS-CoV-2 anti-S Ig values > 300 IU/mL. Stage IV cancer was not found to be significantly associated with an impaired antibody response.

## 3. Discussion

ICPs are generally considered as a homogenous group in terms of vaccine recommendations. In our study, we compared the intensity and durability of the humoral response after primo-vaccination and a booster dose of mRNA vaccine in diverse ICP groups treated in the outpatient consultation of a large tertiary hospital.

We showed that anti-S antibody production after primo-vaccination (two doses) and a booster dose is lower in HSCT patients and LT recipients compared to cancer patients and PLWH. The LT recipient population had a high proportion of poor responders. Furthermore, our analyses revealed that contemporary treatment with immunosuppressive drugs (calcineurin inhibitor, MMF, corticosteroids) used in transplantation, as well as chemotherapy, was associated with lower titers of anti-S antibody after vaccination. Nevertheless, the incidence of COVID-19 after vaccination was found to be similar in ICPs and healthy volunteers and only a minority of transplanted patients required hospitalization and no death related to COVID-19 was reported. The relatively low incidence rate of SARS-CoV-2 infections in ICPs may also be explained by their limited exposure to the virus, together with a higher adherence to preventive recommendations. Notably, prophylactic monoclonal antibody infusion was not available in Switzerland at the time of the study.

Our findings are in line with results from previous studies in different ICP groups. Indeed, many prospective studies have shown that patients with solid tumors had almost similar seroconversion rates (90–95%) after two doses of mRNA vaccine compared to the general population [13,14,15]. However, oncologic treatments received during vaccination have been shown to have an impact on antibody titers, with lower titers for patients on cytotoxic chemotherapy or monoclonal antibodies and with a very low rate of seroconversion, especially in those who received anti-CD20 rituximab in the 6 months prior to vaccination. Hematopoietic malignancies, compared with patients with solid tumors, were less likely to develop a high antibody response [4,5]. In this study, receiving chemotherapy at the time of vaccination was found to be associated with a higher risk of having a “poor” response with an odds ratio (OR) of 0.16 (95% CI: 0.0–0.85). In our analysis, patients with a solid tumor represented 84% of the cancer group and most (81%) had stage IV cancer. However, the cancer stage was not predictive of a poor antibody response (OR: 1.73 [95% CI: 0.75–3.95]). Previous studies have already observed the absence of significant difference in the seroconversion rate among individuals with localized or metastatic tumors [5].

Among PLWH on antiretroviral drugs with a suppressed HIV viral load and high CD4 T-cell count, it was shown that the humoral response following two doses of SARS-CoV-2 mRNA vaccine [16,17] and the booster dose was comparable to the response measured in seronegative healthcare workers [18]. Similarly, we observed that antiretroviral drugs did not impair the humoral immune response to anti-SARS-CoV-2 vaccine. In line with other studies, we also observed an association between higher CD4 levels and being classified as a “good” responder. Corma-Gómez et al. reported CD4 T-cell counts as a major factor with a poorer response of PLWH with CD4 T -cell counts < 200 cells/mm^3^ and with only 64% of seroconversion after two doses compared to 91% in all individuals with HIV [19]. Following the booster dose, the humoral response in individuals with HIV was similar to that measured in healthcare workers [18].

By contrast, transplanted patients (e.g., HSCT and LT patients) showed a reduced humoral immune response compared to all other groups in our study. Indeed, after the first dose, 84% and 38% of LT and HSCT patients, respectively, were poor responders. The third vaccine dose led to a higher proportion of patients reaching > 300 IU/mL in the LT (48%) and HSCT (70%) groups, which confirms the beneficial effect of the booster strategy in this population. These results are consistent with those previously described [9,20]. This reduced capacity to develop antibody responses is mainly due to the deep immune deficiency secondary to immunosuppressive drugs. However, in HSCT patients this may also be due, in some cases, to a worsening of the slow and gradual immune reconstitution after engraftment by immune reaction, such as GVHD.

Several studies have identified potential risk factors for a poor response to vaccines, such as a short time interval from transplantation to vaccine, lymphopenia, contemporary immunosuppressive drugs such as MMF [21,22], or recent treatment with rituximab [23]. We confirmed the negative impact of a low CD4 count (available for HSCT patients) and immunosuppressive drugs at the time of transplantation (prednisone, calcineurin inhibitors, MMF). However, the rate of SARS-CoV-2 infection and its severity was rather low during follow-up in the HSCT and LT populations. We hypothesize that preventive associated measures to avoid viral transmission were followed by these patients who were aware of their increased risk of developing a severe infection. Only a few patients required hospitalization due to COVID-19 during the study (HSCT, 14 group; LT, 3) and none were hospitalized in intensive care, which may be secondary to vaccine efficacy, but also to better management of the COVID infection over time.

There is no uniform SARS-CoV-2 antibody response to mRNA vaccine in the ICP population [11,24]. However, to our knowledge, there is only one prospective cohort study on ICPs with diverse underlying diseases (HIV, solid and hematological malignancies, HSCT, liver and kidney transplantation), which has evaluated the antibody response following two vaccinations with Pfizer-BioNTech vaccine [25]. Rahav et al. showed that younger patients with HIV or solid tumors treated with immunochemotherapy are more likely to develop antibody responses compared to older patients, particularly heart and kidney transplant patients. Underlying immunosuppression and age > 65 years were significantly associated with a non-reactive response of IgG antibodies at 4 weeks following two doses of mRNA COVID-19 vaccine.

Our study has some limitations. First, the healthy individuals were relatives of caregivers at our institution and recruited on a voluntary basis, which may have led to recruitment bias. They were younger (median age, 30 years) and with fewer comorbidities (2%) compared to the overall study population (median age, 56 years; 43% with comorbidities). Second, a diagnosis of SARS-CoV-2 infection was based on a PCR test performed only if the patient was symptomatic or upon detection of new anti-N antibodies positivity during follow-up, thus limiting the accuracy of the exact time of SARS-CoV-2 acquisition. Finally, although the patients recruited were representative of those routinely followed in a large university center, there was an imbalance in the size of the different cohorts. Some characteristics concerning the immunosuppressive status (CD4 counts in the LT group) or long-term follow-up in some groups (M6 after the third dose for cancer and LT patients) were also missing. We also acknowledge that only ICPs willing to receive mRNA SARS-CoV-2 vaccine at the launch of the vaccination campaign in Switzerland were included by the study design. As such, we did not recruit unvaccinated individuals from similar risk groups in a control population. Third, we only assessed the humoral antibody response in our study, whereas the cell-mediated vaccine response has also been shown to be an important determinant of protection [26,27]. Vaccine-induced immunogenicity and the mechanisms that protect against infection, disease or fatal COVID-19 are not yet fully understood or clearly defined.

A T-cell response analysis to SARS-CoV-2 vaccination has been performed in two sub-studies at our university hospital [5,28]. In HSCT-vaccinated patients, the T-cell response was decreased, with a reduced specific T-cell repertoire compared to healthy volunteers [28]. In the second study, which included 131 cancer patients [5], IFN-γ levels were measured to assess the T-cell response. An association was found between the level of antibody response and T-cell response, with 95% of the T-cell response in patients with a high antibody response. By contrast, some patients (44.6%) with no seroconversion after two vaccine doses demonstrated signs of T-cell activation.

Despite these limitations, we consider that the humoral response assessment after anti-SARS-CoV-2 vaccine is easily reproducible and may be a useful tool to provide important information to healthcare workers who look after immunosuppressed patients. Finally, the clinical protective effect of antibody titers induced by vaccination against severe COVID-19 disease should also be balanced by the emergence of new variants in 2021–2022 and the need for enhanced screening and early treatment strategies to prevent severe COVID-19.

In conclusion, this collaborative work confirms the variability of the humoral response after mRNA SARS-CoV-2 vaccine and booster doses between patients with different immunosuppressive conditions, often grouped together as one entity. Future anti-SARS-CoV-2 vaccine recommendations (schedule, booster) should take into account this heterogeneity.

## 4. Methods

### 4.1. Patients

We conducted a prospective, monocentric observational study at Geneva University Hospitals (Geneva, Switzerland) between 1 January 2021 and 31 July 2022. The research was conducted with the approval of the Cantonal Commission for Research Ethics in Geneva (BASEC number, project-ID: Cancer: 2021-00152, HSCT: 2021-01237, HIV: 2021-00491; lung transplant: 2021-00262) in agreement with the amended Declaration of Helsinki. Written informed consent was obtained from each participant prior to inclusion.

Common exclusion criteria were allergic reaction (anaphylaxis) after a previous dose or to a vaccine component, SARS-CoV-2 infection < 3 months, fever ≥ 38 °C; breastfeeding, pregnant or planning to become pregnant, age 18 years and over, and inability to provide informed written consent (due to a language barrier, cognitive disorder, or psychiatric disturbance). Patients with a prior SARS-CoV-2 infection (symptomatic) or exposure (asymptomatic) were not excluded. Asymptomatic exposures to the virus were identified by the presence of positive anti-SARS-CoV-2-S (spike protein) and/or anti-SARS-CoV-2-N (nucleocapsid protein) antibodies. Inclusion in the study was proposed to all patients who received medical care at our hospital in the oncology, infectiology, pneumology and hematology departments and who met the eligibility criteria. The number of patients who declined to receive vaccination or enter the study was not assessed. Healthy individuals were recruited among the relatives of caregivers at our hospital.

#### 4.1.1. Cancer Patients

We included 232 cancer patients. Inclusion criteria included individuals aged 18 years or over who were eligible to receive a COVID-19 vaccination and were diagnosed with any malignancy except for early-stage squamous cell skin cancer, early stage basal cell skin carcinoma and a non-invasive pathology, such as ductal carcinoma in situ. Patients who received anti-cancer treatment within the last 5 years were eligible.

#### 4.1.2. HIV Patients

We included 131 PLWH aged 18 years or over and enrolled in the Swiss HIV Cohort Study at the HIV/AIDS Unit of Geneva University Hospitals.

#### 4.1.3. HSCT Patients

We included 126 patients who received an allogeneic HSCT with the following inclusion criteria: a minimum of 3 months and a maximum of 3 years since allogeneic HSCT or patients transplanted >3 years with GVHD requiring systemic immunosuppressive drugs; absence of rituximab treatment in the previous 3 months; and absence of ongoing steroid treatment of prednisone >10 mg/day or equivalent.

#### 4.1.4. LT Patients

We included 39 LT recipients who were eligible if they received the transplant >3 months previously.

#### 4.1.5. Healthy Individuals

Forty-nine healthy volunteers were recruited between February 2021 and July 2021 within Geneva University Hospitals. Participants were able and willing to provide informed consent and had no contraindications to the SARS-CoV-2 vaccine.

### 4.2. Vaccine Administration and Blood Sample Collection

The vaccination series was administered according to the manufacturer’s guidelines (the delay between the first and second dose was 21 days for mRNA-1273 and 28 days for BNT162b2). A booster dose (also referred to as a third vaccine dose) was administered according to the Swiss Federal Office of Public Health recommendations (https://www.infovac.ch/docs/public/coronavirus/bag/annexe1-COVID-19-recommandation-pour-vaccins-arnm-e--tat-210721.pdf, chapiter 3.2.1 [accessed 20 July 2023]), i.e., from July 2021, only to patients who developed an insufficient antibody response, and then from November 2021 for all patients, irrespective of the antibody response after 2 doses.

Blood samples were collected at 5 time points: (1) at the time of the first vaccine dose (baseline); (2) 1 month after the second dose (M1); (3) from 3 to 5 months after the second dose (M3–M5); (4) 1 month after the booster dose (booster_M1); and (5) 6 months after the booster dose (booster M6). The last time point (booster M6) was collected for HSCT, PLWH and healthy groups only.

### 4.3. Data Collection and SARS-CoV-2 Infection Assessment

Information regarding COVID infection (confirmed by an antigenic or PCR test), symptoms and severity (defined by an COVID infection requiring hospital admission) were assessed at each follow-up visit (day 0 [first vaccine dose], day 28 [second vaccine dose]), i.e., M1 (1 month after the second dose), M3-5 (3–5 months after the second dose), 1 month after the booster dose (M1 booster) and 6 months after the booster dose (M6 booster). Patients who developed a newly detectable anti-N positivity without a history of COVID symptoms were considered to have developed asymptomatic COVID infection.

### 4.4. Anti-SARS-CoV-2 Spike and Nucleocapsid Ig Assays

These samples were tested for both anti-SARS-CoV-2-S Ig and nucleocapsid (N) Ig titers. Blood samples collected using standard sampling tubes were directly centrifuged and serum was stored at −80 °C until batch analysis. The immunogenicity of mRNA vaccines was assessed by the quantitative Elecsys^®^ (Roche Diagnostics, Rotkreuz, Switzerland) anti-S measuring IgA/M/G to the SARS-CoV-2 receptor binding domain. The cut-off value for this assay was 0.8 IU/mL with <0.8 IU/mL values reported as negative, and the maximum value was 2500 IU/mL. This threshold resulted in a sensitivity of 98.8% (95% CI: 98.1–99.3%) in 1610 samples from a cohort of 402 symptomatic patients with PCR-confirmed SARS-CoV-2 infection and a specificity of 99.98% (95% CI: 99.91–100%) in a cohort of 5991 samples from pre-pandemic routine diagnostics and blood donors (Elecsys^®^ Anti-SARS-CoV-2 S. Package Insert, 2020-09, V1.0; Material Numbers 09289267190 and 09289275190). Semi-quantitative Elecsys^®^ anti-N measuring IgA/M/G was used according to the manufacturer’s instructions. The cut-off value for this assay was 1.0.

### 4.5. Statistical Analysis

Continuous variables were reported as means and standard deviations or medians and Q1 and Q3. Categorical variables were reported as counts and percentages. For anti-S antibodies, the GMT were computed together with their 95% CIs. The associations between responding to the vaccine (anti-S antibody values >300 IU/mL) and some candidate predictors were individually tested using generalized estimating equations to account for repeated measures. Healthy individuals and individuals identified as having already been infected with COVID at baseline were excluded from these analyses. ORs with their 95% CIs were reported. ORs > 1 indicate a higher chance of being a good responder (anti-S > 300). Analyses and graphs were created using R software (v.4.2.2; R).

## Figures and Tables

**Figure 1 vaccines-11-01284-f001:**
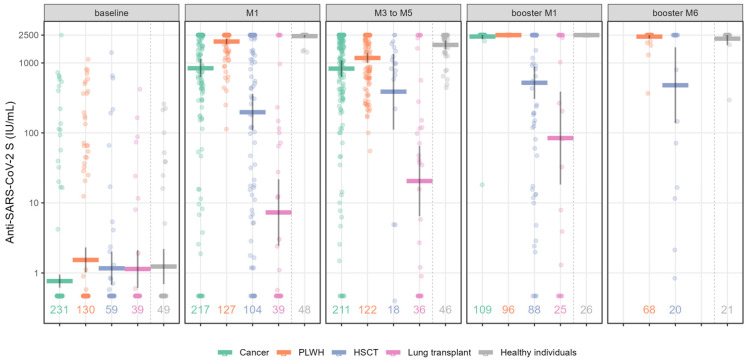
Quantification of anti-SARS-CoV-2 S (anti-S) Ig titers following complete mRNA vaccination and booster dose at different time points in immunocompromised patients and healthy individuals. Values along the x axis represent the number of individuals in the respective group at each time point. Scatterplots show anti-S values as geometrical mean titers with their 95% CIs. Numbers represent the number of patients per group and per time point. Abbreviations: HSCT: hematopoietic stem cell transplantation; PLWH: people living with HIV.

**Figure 2 vaccines-11-01284-f002:**
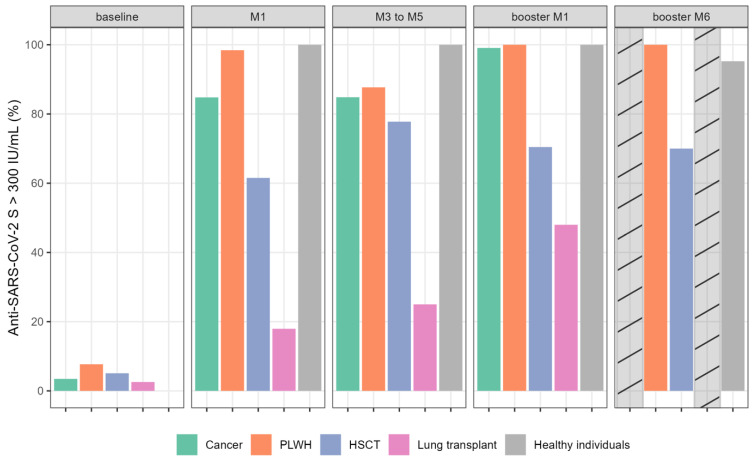
Proportion of immunocompromised patients and healthy individuals with anti-SARS-CoV-2 S (anti-S) Ig titers > 300 IU/mL (%) following complete mRNA vaccination and booster dose at different time points. Bar plots show the percentage of patients with anti-S > 300 IU/mL per group per time point. Cross-hatched bars indicate missing data. Abbreviations: HSCT: hematopoietic stem cell transplantation; PLWH: people living with HIV.

**Figure 3 vaccines-11-01284-f003:**
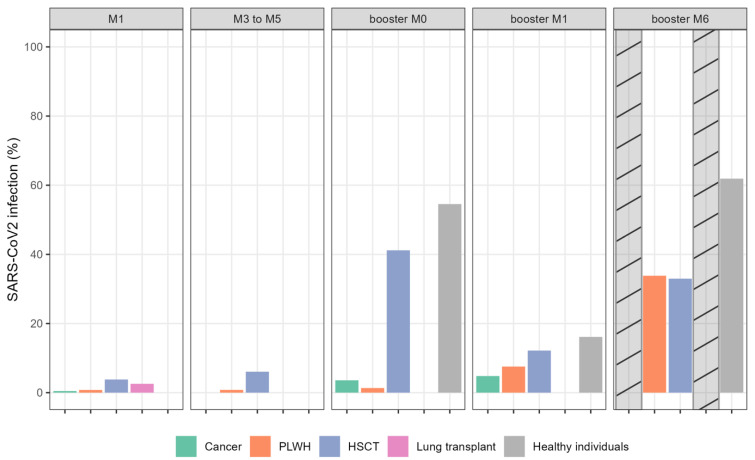
Proportion of patients with SARS-CoV-2 infection (%). Barplots showing percentage of COVID infection per group per timepoint. The cross-hatched bars indicate missing data. Abbreviations: HSCT: hematopoietic stem cell transplantation; PLWH: people living with HIV.

**Table 1 vaccines-11-01284-t001:** Clinical characteristics of the study cohort.

Variables	Total (*n* = 577)	Cancer (*n* = 232 [40%])	PLWH (*n* = 131 [23%])	HSCT (*n* = 126 [22%])	LT Recipients (*n* = 39 [7%])	Healthy Individuals (*n* = 49 [8%])
Age (years; median [Q1–Q3])	56 (45–66)	61 (50.8–69)	54 (47–60.5)	59 (47.2–67)	59 (51.5–67)	30 (27–34)
≥ 65 years old	143 (25%)	85 (37%)	9 (7%)	34 (27%)	15 (38%)	0
Sex						
Female	236 (41%)	111 (48%)	40 (31%)	44 (35%)	21 (54%)	20 (41%)
Male	341 (59%)	121 (52%)	91 (69%)	82 (65%)	18 (46%)	29 (59%)
Comorbidity						
Yes	250 (43%)	106 (46%)	48 (37%)	68 (54%)	27 (69%)	1 (2%)
Hypertension	170 (29%)	66 (28%)	35 (27%)	46 (37%)	22 (56%)	1 (2%)
Diabetes	64 (11%)	25 (11%)	14 (11%)	19 (15%)	6 (15%)	0
Obesity	29 (11%)	25 (11%)	-	-	4 (10%)	-
Heart disease	59 (10%)	17 (7%)	17 (13%)	24 (19%)	1 (3%)	0
Tobacco use	27 (10%)	25 (11%)	-	-	2 (5%)	-
COPD	55 (10%)	12 (5%)	9 (7%)	34 (27%)	0	0

Abbreviations: Q: quartiles; -: data not collected; HSCT: hematopoietic stem cell transplantation; PLWH: people living with HIV; COPD: chronic obstructive pulmonary disease.

**Table 2 vaccines-11-01284-t002:** Geometric mean titer for anti-SARS-CoV-2 (anti-S) Ig.

	Geometric Mean Titer (95% CI) at Different Time Points				
	Baseline	M1	M3 to M5	Booster M1	Booster M6
Cancer	0.8 [0.6–0.9] (*n* = 231)	840 [625–1129] (*n* = 217)	830 [636–1082] (*n* = 211)	2386 [2182–2500] (*n* = 109)	-
PLWH	1.5 [1.0–2.3] (*n* = 130)	2024 [1854–2209] (*n* = 127)	1177 [1003–1380] (*n* = 122)	2500 [-] (*n* = 96)	2379 [2240–2500] (*n* = 68)
HSCT	1.2 [0.7–2.0] (*n* = 59)	198 [108–361] (*n* = 104)	388 [111–1358] (*n* = 18)	521 [306–885] (*n* = 88)	480 [137–1676] (*n* = 20)
Lung transplant	1.1 [0.6–2.1] (*n* = 39)	7.3 [2.5–22] (*n* = 39)	20 [6.5–65] (*n* = 36)	84 [18–389] (*n* = 25)	-
Healthy individuals	1.2 [0.7–2.2] (*n* = 49)	2417 [2327–2500] (*n* = 48)	1812 [1570–2092] (*n* = 46)	2500 [-] (*n* = 26)	2224 [1797–2500] (*n* = 21)

Abbreviations: M: month; HSCT: hematopoietic stem cell transplantation; PLWH: people living with HIV.

**Table 3 vaccines-11-01284-t003:** Associations between different predictors and anti-SARS-CoV-2 S Ig titers >300 IU/mL.

Predictors	Odds Ratio	95% CI
Antiretroviral treatment	6.57	3.96–10.88
Number of CD4 (by 100 units)	1.28	1.12–1.47
Year from transplant	1.13	0.79–1.61
Prednisone	0.13	0.03–0.49
Tacrolimus	0.23	0.07–0.72
Mycophenolate mofetil	0.05	0.01–0.41
Chemotherapy	0.16	0.03–0.85
Cancer stage IV	1.73	0.75–3.95

## Data Availability

The datasets used and/or analyzed during the current study are available from the corresponding author on reasonable request.

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
