# Peer review of "SARS-CoV-2 m-RNA Vaccine Response in Immunocompromised Patients: A Monocentric Study Comparing Cancer, People Living with HIV, Hematopoietic Stem Cell Transplant Patients and Lung Transplant Recipients"

_vaccines, 2023, doi:10.3390/vaccines11081284_

Round 1

Reviewer 1 Report

This study described the variability of humoral response following receiving mRNA COVID-19 vaccine and booster dose between patients with immunosuppressive conditions and healthy controls. It would provide very preliminary evidence for understanding the COVID-19 vaccination among the patients with immunosuppressive conditions. Several comments are as follows:

1, This study included patients with immunosuppressive conditions, who had all received the COVID-19 vaccination. However, I wonder if the study could recruit the patients with similar immunosuppressive conditions who had natural infection of SARS-CoV-2 and/or did not receive COVID-19 vaccines. It may answer more questions, such as the hypothesis "humoral response to SARS-CoV-2 might be weak among the patients with immunosuppressive conditions", regardless of natural infection or vaccination.

2, I suggest Table 2 be reframed, such as "Group" X "Points of time", in which GMT could be presented in each cell.

3, In Figure 3, SARS-CoV-2 infection is presented. However, considering the possibility of asymptomatic infection (particularly Omicron variants in the end of 2021 through 2022), the infection and diagnosis might not be reliable. Furthermore, temporality between the COVID-19 vaccination and natural infection could not be determined.

4, Also, limited sample size and representativenee of the study should be presented in the study limitations, as the findings in the study might have limited generalizability in this hospital as well as to larger groups of patients with immunosuppressive conditions.

Author Response

This study described the variability of humoral response following receiving mRNA COVID-19 vaccine and booster dose between patients with immunosuppressive conditions and healthy controls. It would provide very preliminary evidence for understanding the COVID-19 vaccination among the patients with immunosuppressive conditions. Several comments are as follows:
Point 1-This study included patients with immunosuppressive conditions, who had all received the COVID-19 vaccination. However, I wonder if the study could recruit the patients with similar immunosuppressive conditions who had natural infection of SARS-CoV-2 and/or did not receive COVID-19 vaccines. It may answer more questions, such as the hypothesis "humoral response to SARS-CoV-2 might be weak among the patients with immunosuppressive conditions", regardless of natural infection or vaccination.
Our study focuses on the patients with immunosuppressive conditions who were eligible to priority vaccination against COVID when it became available. As a result, we deliberately recruited patients eligible to receive a COVID-19 vaccine at the time of the launch of the vaccination campaign. As part of the study design, we did not recruit patients unwilling to be vaccinated. 
However, another study made by our team on hematopoietic stem cell transplanted patients, recruited patient who have contracted COVID before vaccination. We concluded that humoral response and T cell response (TCR repertoire diversity) was weaker after COVID infection as well as after vaccination in this population, compared with healthy controls. (Pradier A, Mamez AC et al. T cell receptor sequencing reveals reduced clonal breadth of T-cell responses against SARS-CoV-2 after natural infection and vaccination in allogeneic hematopoietic stem cell transplant recipients. Ann Oncol. 2022 Dec;33(12):1333-1335.).

Point 2- I suggest Table 2 be reframed, such as "Group" X "Points of time", in which GMT could be presented in each cell.
We have reframed the Table 2 and added the corrected version in the manuscript.

Point 3- In Figure 3, SARS-CoV-2 infection is presented. However, considering the possibility of asymptomatic infection (particularly Omicron variants in the end of 2021 through 2022), the infection and diagnosis might not be reliable. Furthermore, temporality between the COVID-19 vaccination and natural infection could not be determined.
Indeed, as mentioned in the methods (lines 416-422), we choose to include the outcome SARS Cov2 infection whereas it was diagnosed with standard PCR test or when we detect during the follow up a new positivity in anti-N antibody detection. The proportion of patient with PCR-based COVID diagnosis was 62% compared to 38% on when the diagnosis was solely based on the occurrence of the detection of Anti-N positivity.
Our goal was to be exhaustive, but it brings limitation regarding the timing of infection from the vaccination. We have added details and text to highlight this limitation in the manuscript:
-page 3, Section: Results, paragraph “Proportion of immunocompromised patients with SARS-COV-2 infection “, lines 189-192: “Out of all COVID diagnosis, 62% were identifies using a PCR test and 38% were based on new detection of Anti-N antibody during follow-up, revealing asymptomatic or poorly symptomatic SARS-CoV-2 infection. SARS-CoV-2 variant determination was not performed.”

-page 9, Section: Discussion, lines 326-328: “Diagnosis of SARS Cov2 infection was based on PCR testing when patient was symptomatic or detection of new positivity of anti-N antibodies detection during the follow up, limiting the accuracy in the temporality of the infection in relation to the vaccination time. »

Point 4- Also, limited sample size and representativeness of the study should be presented in the study limitations, as the findings in the study might have limited generalizability in this hospital as well as to larger groups of patients with immunosuppressive conditions.
We have highlighted this point in the manuscript:
Page 9, Section: Discussion, lines 329-331:« Moreover, whereas the recruited patient represented our university regional center immunosuppressed population, group sizes are inequal and sample size are reduced in some population as lung transplant group “.

Reviewer 2 Report

It would be good to present in the abstract the most critical results founding  the study's final conclusion. Describing a response as poor is not sufficient even in the abstract section.

Again in the introduction section, more detailed information on the previous ICP vaccine response analysis should be provided.

The method employed to approach patients for recruitment should be described including the way of patients' invitations and  the proportion of those declining.

The groups of the patients are not uniform, to exclude the bias behind conclusions I would propose to make statistical analysis to control confounding effects due to the variety of symptomatology in the patients presented under the same name

In cancer pts: how are solid tumor patients at stage IV vs the rest?

HIV patients respond positively to the treatment against those evolving to the disease.

Providing more details depicting the critical parameters associated with vaccination response would be appreciated by the readers.

Anti Ig S titer is not a sufficient description, especially since anti-nucleocapsid antibodies were examined, as well.

What was the variant of SARS CoV 2 causing more recent infections? If identified it should be discussed.

The text is a bit wordy - see my recommendations

Author Response

Point 1 -It would be good to present in the abstract the most critical results founding the study's final conclusion. Describing a response as poor is not sufficient even in the abstract section.

We have added more detailed results in the Abstract section (lines 34-41):“ After primo-vaccination, the highest anti-S antibody GMT (IU/ml) was observed in healthy individuals (2417 IU/ml [95%CI 2327 - 2500]), PLWH group (2024 IU/ml [95%CI 1854 - 2209]), patients with cancer (840 IU/ml [95%CI 625 -1129], whereas HSCT group and Lung transplant group patients had the weaker antibody response (198 IU/ml [95%CI 108 - 361] and 7.3 IU/ml  [95%CI 2.5 - 22]. Booster dose conferred a high antibody response after 1 month in both PLWH (2500 IU/ml) and in individuals with a cancer (2386 IU/ml [95%CI 2182 - 2500], a moderate response in HSCT patients (521 IU/ml [95%CI 306 - 885] and a poor response in lung transplant patients (84 IU/ml [95%CI 18 - 389]. “

Point 2- Again, in the introduction section, more detailed information on the previous ICP vaccine response analysis should be provided.

As suggested, we added more detailed information in the introduction section about published data on response to SARS-COv-2 vaccine:

Page 2, Introduction, Lines 57-66:

“ Compared to general population, people living with HIV (PLWH) and cancer patients with solid tumors showed similar seroconversion rates (90-95%) after two and three doses of mRNA vaccines. However, oncologic treatments received during the vaccination showed to have an impact on antibody titers, with lowest for patients treated with cyto-toxic chemotherapy or monoclonal antibody. In the other side, allogeneic Hematopoietic Stem cell (HSCT) and lung transplant (LT) recipients were less likely to develop humoral immune response after mRNA vaccines. A recent meta-analysis highlighted a poor seroconversion rate after covid-19 vaccination in transplanted patient with only one third achieving seroconversion after 2 doses.”

Point 3- The method employed to approach patients for recruitment should be described including the way of patients' invitations and the proportion of those declining.

Our study has been set up at the time SARS COv2 vaccination was launched in Switzerland, with a priority given to immunocompromised patients. The study has been proposed to all eligible patients routinely followed in our hospital. We did not collect data regarding patients who declined vaccination or refused to participate to this observational study.

We have added some detailed on this point in the result part of the manuscript: Page 10, Section: Methods, lines 373-377:

“Inclusion in the study has been proposed to all patients who received medical care in our hospital in Oncology department, Infectiology department, Pneumology department or Hematology department, and who met the eligibility criteria. The number of patients who decline to receive vaccination or to enter the study has not been assessed.”

Point 4-The groups of the patients are not uniform, to exclude the bias behind conclusions I would propose to make statistical analysis to control confounding effects due to the variety of symptomatology in the patients presented under the same name.

In cancer pts: how are solid tumor patients at stage IV vs the rest?

HIV patients respond positively to the treatment against those evolving to the disease.

Indeed, in each group, we recognize a certain heterogeneity, which reflect routine practice in a University Hospital such as ours.  In Cancer group, 84% had a solid tumor (versus 16% with hematologic malignancies) and 60% of solid tumor group had a stage IV cancer. In PLWH, 85% had an indetectable viral load at the time of first dose vaccine.

To better analyze subgroups within this heterogeneous population, we included these candidate predictors within a previously done analysis (Results were added in Table 3).  

Having a stage 4 cancer was not identified has a significant predicting factor associated with poor antibody response (Odds ratio 1.73, 95% confidence interval 0.75 to 3.95).

To note, in PLWV group, we decided not to include a sensitivity analysis based on the detectable or undetectable HIV-RNA load, or based on CD4 cells; indeed, the number of patients with a poor antibody response (less than 300UI/L) was low and did not allow for such sensitivity analyses.

We have now added:

-Revised Results in Table 3 (Page 8, Section: Results, line 234):

Table 3. Associations between different predictors and anti-SARS-COV-2 S Ig titers >300 IU/mL

Predictors

Odds ratio

95% confidence interval

Antiretroviral treatment

6.57

3.96 to 10.88

Number of CD4 (by 100 units)

1.28

1.12 to 1.47

Year from transplant

1.13

0.79 to 1.61

Prednisone

0.13

0.03 to 0.49

Tacrolimus

0.23

0.07 to 0.72

Mycophenolate mofetil

0.05

0.01 to 0.41

Chemotherapy

0.16

0.03 to 0.85

Cancer stage V (versus I,II,III)

1.73

0.75 to 3.95

-Changes in the statistical analysis part (page 12, paragraph: Statistical analysis, lines 442-446):

“The associations between responding to the vaccine (having anti-S antibodies values above 300) and some candidate predictors were individually tested using Generalized Estimating Equations (GEE) to account for the repeated measures”

-Description in the result section (page 7, section: results, paragraph: Response factors according to immunosuppression group, lines 208-225):

“Response factors according to immunosuppression group

Finally, we tested the association between different predictors (e.g., immunosuppressive treatments, immune status of patients, stage of solid tumor) and anti-SARS-CoV-2 anti-S Ig values above 300 IU/ml at any timepoint followed vaccination (“strong responder”). In order to limit the scope of the analyses to immunosuppressed patients, healthy volunteers were excluded from those analyses. Moreover, patients who already reached an-ti-SARS-CoV-2 anti-S Ig values above 300 IU/ml at baseline were also excluded. The odds ratios and their 95% confidence intervals are reported in Table 3. Immunosuppressive treatments during vaccination for in transplanted population (HSCT and Lung transplanted patients) such as prednisone, tacrolimus, MMF and chemotherapy were inversely associated with anti-SARS-CoV-2 anti-S Ig values above 300 IU/ml at any timepoint after vaccination, meaning that patients receiving these treatments were less likely to develop a “strong response”.

CD4 counts was measured in PLWH and HSCT patients at the time of first vaccine. Average CD4 T-cell count: 646.9 (±294.5) cells/mm3 and 226 (111-423) cells/mm3 respectively. CD4 count were positively associated with anti-SARS-CoV-2 anti-S Ig values above 300 IU/ml.

 Having a stage IV cancer was not found to be statistically significantly associated with antibody response.”

Point 5-Providing more details depicting the critical parameters associated with vaccination response would be appreciated by the readers.

We have added some detailed on the critical parameters associated with response to vaccination:

Page 7, Section: Results, paragraph: Response factors according to immunosuppression group, lines 208-225):

« Response factors according to immunosuppression group

Finally, we tested the association between different predictors (e.g., immunosuppressive treatments, immune status of patients, stage of solid tumor) and anti-SARS-CoV-2 anti-S Ig values above 300 IU/ml at any timepoint followed vaccination (“strong responder”). In order to limit the scope of the analyses to immunosuppressed patients, healthy volunteers were excluded from those analyses. Moreover, patients who already reached an-ti-SARS-CoV-2 anti-S Ig values above 300 IU/ml at baseline were also excluded. The odds ratios and their 95% confidence intervals are reported in Table 3.

Immunosuppressive treatments during vaccination for in transplanted population (HSCT and Lung transplanted patients) such as prednisone, tacrolimus, MMF and chemotherapy were inversely associated with anti-SARS-CoV-2 anti-S Ig values above 300 IU/ml at any timepoint after vaccination, meaning that patients receiving these treatments were less likely to develop a “strong response”.

CD4 counts was measured in PLWH and HSCT patients at the time of first vaccine. Average CD4 T-cell count: 646.9 (±294.5) cells/mm3 and 226 (111-423) cells/mm3 respectively. CD4 count were positively associated with anti-SARS-CoV-2 anti-S Ig values above 300 IU/ml.

 Having a stage IV cancer was not found to be statistically significantly associated with antibody response.”

Point 6-Anti Ig S titer is not a sufficient description, especially since anti-nucleocapsid antibodies were examined, as well.

We selected Anti-S antibody titer as a primary endpoint to study vaccine responses, based on the validated and available laboratory routine practices. Anti-N measured was assessed contemporary to IgG (same blood sample) and was not considered to reflect vaccine humoral response but rather as a marker of a new COVID-19 infection.

Neutralizing antibodies measured would have been of interest but was not validated nor available in our center. Finally, the measure of Anti S antibodies was standardized, reproducible which allow comparison with other studies performed in different context. Indeed, in several countries’ guidelines, antibody anti-S titers were considered to be a sufficiently good surrogate of vaccine response to inform the decision of providing a third dose or an antibody-based prophylaxis.

Point 7-What was the variant of SARS CoV 2 causing more recent infections? If identified, it should be discussed.

Information’s regarding the variants causing COVID infection was not available. However, looking at the repartition of COVID infection during time in parallel with local epidemiology, we can observe that a majority of COVID infection occurred when OMICRON variants became the predominant variant.

Geneva epidemiology

DELTA wave

OMICRON wave

Time

Jan-March 2021

April-Aug 2021

Sept - Dec

 2021

Jan - Feb 2022

March 2022- July 2022

COVID-19 cases

14%

3%

14%

43%

26%

We added more detailed on the period of infection within the manuscript:

Page 6, section: Results, paragraph “Proportion of immunocompromised patients with SARS-COV-2 infection “, lines 192-19: “A majority of COVID infection (69%) were diagnosed after January 2022, while Omicron variants became predominant in our local SARS-Cov-2 epidemiology”

Reviewer 3 Report

The manuscript by Bordry et al consists of collaborative monocentric prospective cohort study, in which the authors assessed the anti-SARS-CoV- spike protein antibody titers following  two and three doses of mRNA vaccines in four groups of Immunocompromised patients  (cancer, hematopoietic stem cell  transplantation(HSCT), people living with HIV (PLWH) and lung transplant (LT recipients)) treated  in the Geneva University Hospitals as well as in healthy individuals. Anti-S antibody production after primo-vaccination, booster dose was lower in HSCT and LT patients than those with cancer patients, PLWH and healthy individuals. Contemporary treatment with immunosuppressive drugs used in transplantation or chemotherapy was associated with poor response to vaccination.

Though the manuscript is well-written, there is need to address the following concern before considering it for publication.

1.      The median age of healthy individuals was 30, while for other groups the median age was above 50. The studies conducted in the past suggest that SARS-CoV-2 viral specific antibody response profiles are distinct in different age groups. Therefore the authors are requested justify this significant age differences in the control group compared to ICP groups.

2.      Line 92 to 99: What was the average CD4 cell count in LT group.  As per table 3, CD4 count as a predictors has a positive correlation with strong antibody responses.

3.      Figure 1: What does the values at x axis denote. Though it can be understood as number of subjects in the respective group at a particular time point, but it should be mentioned in the Figure or its legend for better clarity.

4.      Why antibody was not measured at M6 after booster dose in Cancer and LT groups? It is important to assess the follow up data for all group in order to determine the duration of protection provided by the vaccine in the different immunocompromised groups.

5.      The authors should have assessed the T cell responses in the different groups. Various studies suggest that SARS-CoV-2-specific T cell responses are essential for viral clearance, may prevent infection without seroconversion.

6.      Please correct the year as “2023” in the header section.

Minor improvement in language and proof reading of manuscript is recommended.

Author Response

Reviewer 3

The manuscript by Bordry et al consists of collaborative monocentric prospective cohort study, in which the authors assessed the anti-SARS-CoV- spike protein antibody titers following  two and three doses of mRNA vaccines in four groups of Immunocompromised patients  (cancer, hematopoietic stem cell  transplantation(HSCT), people living with HIV (PLWH) and lung transplant (LT recipients)) treated  in the Geneva University Hospitals as well as in healthy individuals. Anti-S antibody production after primo-vaccination, booster dose was lower in HSCT and LT patients than those with cancer patients, PLWH and healthy individuals. Contemporary treatment with immunosuppressive drugs used in transplantation or chemotherapy was associated with poor response to vaccination.

Though the manuscript is well-written, there is need to address the following concern before considering it for publication.

Point 1- The median age of healthy individuals was 30, while for other groups the median age was above 50. The studies conducted in the past suggest that SARS-CoV-2 viral specific antibody response profiles are distinct in different age groups. Therefore, the authors are requested justify this significant age differences in the control group compared to ICP groups.

We agree with the reviewer that the healthy individual group is significantly younger compared with the patient groups; any meaningful comparison with the population of immune-suppressed patients is therefore difficult and should be interpreted with caution.

These control individuals were recruited in the relatives of caregivers of our hospital at a time when vaccination was not yet available for general young population. This explains their younger age, as the older or comorbid individuals had already been able to receive the COVID-19 vaccine following the Swiss national recommendations at this time.

We have highlighted these limitations in the manuscript:

Page 9, Section: Discussion, lines 319-325.

“First, the healthy individuals, considered as a control group, were recruited in the relatives of caregivers of our hospital and were younger with a median age of 30 years old, and with fewer comorbidities (2%), compared to entire population (median age 56 years old and 43% with comorbidities). Furthermore, they were recruited on a volunteer basis, which induce recruitment bias. This bias limit the comparison between this healthy groups and immunocompromised patients.”

Point 2- Line 92 to 99: What was the average CD4 cell count in LT group.  As per table 3, CD4 count as a predictors has a positive correlation with strong antibody responses.

Unfortunately, CD4 cell counts were not routinely recorded in the lung transplanted population. We added a highlight regarding this point within the text:

Page 10, Section: Discussion, lines 331-332: « Some characteristics relative to immunosuppressive status (as CD4 counts in lung transplant group)

Point 3- Figure 1: What does the values at x axis denote. Though it can be understood as number of subjects in the respective group at a particular time point, but it should be mentioned in the Figure or its legend for better clarity.

Indeed, in Figure 1, x axis denote the number of subject in each group. We have now modified the Legend of Figure 1 to clarify this point (Page 4, lines 150-151)

Point 4- Why antibody was not measured at M6 after booster dose in Cancer and LT groups? It is important to assess the follow up data for all group in order to determine the duration of protection provided by the vaccine in the different immunocompromised groups.

As pointed by the reviewer, long term (M6) antibody titers follow up is missing in the Cancer and Lung transplant groups. Conclusions about the duration of vaccine protection after booster can only be made for PLWV and HSCT in this study. We have now added some explanations regarding this missing value in the limitations:

Page 10, Section: Discussion, lines 331-333:

“Some characteristics relative to immunosuppressive status (as CD4 counts in lung transplant group) or long-term follow in some groups (M6 after third dose for Cancer and Lung transplants patients) are missing.”

Point 4- The authors should have assessed the T cell responses in the different groups. Various studies suggest that SARS-CoV-2-specific T cell responses are essential for viral clearance, may prevent infection without seroconversion.

We agree that the T cell response assessment would have been of a great value in this study. We mentioned this point in the limitations. Unfortunately, the testing of T cell response was, at this time, not routinely accessible in our University Hospital.

To note, sub studies have been done among two of the groups presented in this manuscript. In HSCT patients, T cell response assessed with IFN gamma was shown decreased in HSCT vaccinated patients compared to healthy volunteers and TCR sequencing reveals reduced breath of T cell diversity in HSCT population. A second study showed that, in Cancer population, a correlation was found between the level of antibody response and T cell response (assessed with the measure of IFN gamma level) but that some patients with no humoral response had developed a T cell response.

We agree with these limitations, but we think that the assessment of humoral response is easily available and reproducible and will bring a useful information to the health care workers in charge of immune-suppressed patients.

We added details in the text:

Page 10, Section: Discussion, lines 334-349:

“Finally, we only assessed the humoral antibody response in our study whereas the cell-mediated vaccine response has been showed to be also an important determinant of protection. Vaccine-induced immunogenicity and the mechanisms that protect against infection, disease or fatal COVID-19 are not yet fully understood nor clearly defined.

The analysis of T-cell response to SARS-CoV-2 vaccination has been performed in two sub studies performed in our University Hospital (ref 5,28). In HSCT patients, T cell response assessed with ELISpot was shown decreased in HSCT vaccinated patients compared to healthy volunteers and TCR sequencing reveals reduced clonal breath of T cell response in HSCT population (ref 28). In the second study including 131 patients with cancer (ref 5) , IFN-γ levels was measured to assess T cell response to vaccination. An association was found between the level of antibody response and T cell response with 95% of T cell response in patient with high anti body response. In the other hand, some patients (44.6%) with no seroconversion after 2 doses of vaccine showed a T cell activation.

Despite this limitation, we think that the assessment of humoral response is easily available and reproducible and can bring a useful information to the health care workers in charge of immune-suppressed patients.”

References:

5          Bordry, N. et al. Humoral and cellular immunogenicity two months after SARS-CoV-2 messenger RNA vaccines in patients with cancer. iScience 25, 103699, doi:10.1016/j.isci.2021.103699 (2022).

28        Pradier, A. et al. T cell receptor sequencing reveals reduced clonal breadth of T-cell responses against SARS-CoV-2 after natural infection and vaccination in allogeneic hematopoietic stem cell transplant recipients. Ann Oncol 33, 1333-1335, doi:10.1016/j.annonc.2022.09.153 (2022).

Point 5-Please correct the year as “2023” in the header section.

This has been corrected.

Round 2

Reviewer 1 Report

I suggest the authors might add the response to point 1 in the discussion section.

Author Response

As suggested by the reviewer we added the response to point 1 in the discussion part (line 336 to 339

Reviewer 3 Report

Though there are limitations in the study as agreed by the authors but same has now been highlighted in the manuscript and same can now be accepted for publication.

Author Response

We thank the reviewer for his/her comment